# Highlights on the Development, Related Patents, and Prospects of Lenacapavir: The First-in-Class HIV-1 Capsid Inhibitor for the Treatment of Multi-Drug-Resistant HIV-1 Infection

**DOI:** 10.3390/medicina59061041

**Published:** 2023-05-28

**Authors:** Tafadzwa Dzinamarira, Mazen Almehmadi, Ahad Amer Alsaiari, Mamdouh Allahyani, Abdulelah Aljuaid, Abdulaziz Alsharif, Abida Khan, Mehnaz Kamal, Ali A. Rabaan, Amal H. Alfaraj, Bashayer M. AlShehail, Nouf Alotaibi, Shams M. AlShehail, Mohd Imran

**Affiliations:** 1School of Health Systems and Public Health, University of Pretoria, Pretoria 0002, South Africa; 2ICAP, Columbia University, Harare P.O. Box 28, Zimbabwe; 3Department of Clinical Laboratory Sciences, College of Applied Medical Sciences, Taif University, Taif 21944, Saudi Arabia; 4Department of Pharmaceutical Chemistry, Faculty of Pharmacy, Northern Border University, Rafha 91911, Saudi Arabia; aqua_abkhan@yahoo.com; 5Department of Pharmaceutical Chemistry, College of Pharmacy, Prince Sattam Bin Abdulaziz University, Al-Kharj 11942, Saudi Arabia; 6Molecular Diagnostic Laboratory, Johns Hopkins Aramco Healthcare, Dhahran 31311, Saudi Arabia; 7College of Medicine, Alfaisal University, Riyadh 11533, Saudi Arabia; 8Department of Public Health and Nutrition, The University of Haripur, Haripur 22610, Pakistan; 9Pediatric Department, Abqaiq General Hospital, First Eastern Health Cluster, Abqaiq 33261, Saudi Arabia; 10Pharmacy Practice Department, College of Clinical Pharmacy, Imam Abdulrahman Bin Faisal University, Dammam 31441, Saudi Arabia; 11Clinical Pharmacy Department, College of Pharmacy, Umm Al-Qura University, Makkah 21955, Saudi Arabia; 12Internal Medicine Department, King Abdulaziz University Hospital, King Abdulaziz University, Jeddah 21487, Saudi Arabia

**Keywords:** Lenacapavir, Sunlenca, GS-6207, HIV-1 capsid, patent, prospects

## Abstract

The multidrug-resistant (MDR) human immunodeficiency virus 1 (HIV-1) infection is an unmet medical need. HIV-1 capsid plays an important role at different stages of the HIV-1 replication cycle and is an attractive drug target for developing therapies against MDR HIV-1 infection. Lenacapavir (LEN) is the first-in-class HIV-1 capsid inhibitor approved by the USFDA, EMA, and Health Canada for treating MDR HIV-1 infection. This article highlights the development, pharmaceutical aspects, clinical studies, patent literature, and future directions on LEN-based therapies. The literature for this review was collected from PubMed, authentic websites (USFDA, EMA, Health Canada, Gilead, and NIH), and the free patent database (Espacenet, USPTO, and Patent scope). LEN has been developed by Gilead and is marketed as Sunlenca (tablet and subcutaneous injection). The long-acting and patient-compliant LEN demonstrated a low level of drug-related mutations, is active against MDR HIV-1 infection, and does not reveal cross-resistance to other anti-HIV drugs. LEN is also an excellent drug for patients having difficult or limited access to healthcare facilities. The literature has established additive/synergistic effects of combining LEN with rilpivirine, cabotegravir, islatravir, bictegravir, and tenofovir. HIV-1 infection may be accompanied by opportunistic infections such as tuberculosis (TB). The associated diseases make HIV treatment complex and warrant drug interaction studies (drug–drug, drug–food, and drug–disease interaction). Many inventions on different aspects of LEN have been claimed in patent literature. However, there is a great scope for developing more inventions related to the drug combination of LEN with anti-HIV/anti-TB drugs in a single dosage form, new formulations, and methods of treating HIV and TB co-infection. Additional research may provide more LEN-based treatments with favorable pharmacokinetic parameters for MDR HIV-1 infections and associated opportunistic infections such as TB.

## 1. Introduction

Human immunodeficiency virus (HIV) infection is one of the important public health concerns. HIV-1 and HIV-2 are two genetically distinct types of HIV, wherein the global prevalence of HIV-1 infection (about 95%) is more than HIV-2 [1]. Worldwide, approximately 38.4 million people are afflicted with HIV-1 infection, with approximately 28.7 million receiving antiretroviral therapy (ART) [2]. The HIV-1 infection weakens the immune system of infected people and may progress to acquired immunodeficiency syndrome (AIDS). AIDS is characterized by a CD4+ T-cell count of <200 cells per microliter or HIV infection with associated diseases (tuberculosis, cryptococcal meningitis, lymphomas, Kaposi’s sarcoma, etc.). The increased viral load of HIV-1 and decreased CD4+ T-cell count are the principal markers of HIV-1 infection progression or treatment failure [3]. When left untreated, HIV-1 infection or AIDS can also lead to death after a latency period of various lengths. The HIV-1 infection can be managed but not cured by ART regimens/classes (Figure 1) [4,5,6,7,8].

The increased use of ART has been supplemented by the development of drug resistance, including multi-drug resistance (MDR), in recent years. The unaddressed drug resistance threatens the efficacy of ART and can lead to a rise in new HIV-1 infections and HIV-related morbidity and mortality [9]. The MDR HIV-1 infection (unmet medical need) is refractory to current ART and demands the development of new treatments [10]. Recently, the United States Food and Drug Administration (USFDA). Health Canada and the European Medicines Agency (EMA) have approved Lenacapavir (LEN) as the first-in-class capsid inhibitor for the treatment of MDR HIV-1 infection [11,12,13,14]. Some reviews on LEN have been published, but they are silent about the pharmaceutical aspects, inventions, and patent literature of LEN [11,12,13,15,16,17,18,19]. This review highlights the pharmaceutical aspects, development, related patents, and prospects for LEN. The literature for this review was collected from PubMed, authentic websites (USFDA, Health Canada, EMA, Gilead, and NIH), and free patent database (Espacenet, USPTO, and Patent scope) employing selected keywords (GS-6207, GS-714207, Lenacapavir, Sunlenca, and J05-AX31) or their combinations.

## 2. Lenacapavir (LEN)

### 2.1. Description

LEN (Synonyms: Sunlenca, GS-6207, GS-714207, GS-CA1, and J05-AX31; Molecular Formula: C_39_H_32_ClF_10_N_7_O_5_S_2_; Molecular Weight: 968; CAS registry number: 2189684-44-2) is a weakly acidic indazole derivative with low water solubility and permeability (BCS class 4) [12,20,21]. LEN has been approved in Europe, Canada, and the USA as Sunlenca (Table 1) [22,23,24].

Lenacapavir sodium (Molecular Formula: C_39_H_31_ClF_10_N_7_NaO_5_S_2_; Molecular Weight: 990; Chemical Name: sodium (4-chloro-7-(2-((S)-1-(2-((3bS,4aR)-5,5-difluoro-3-(trifluoromethyl)-3b,4,4a,5-tetrahydro-1H-cyclopropa[3,4]cyclopenta[1,2-c]pyrazol-1-yl)acetamido)-2-(3,5-difluorophenyl)ethyl)-6-(3-methyl-3-(methylsulfonyl)but-1-yn-1-yl)pyridin-3-yl)-1-(2,2,2-trifluoroethyl)-1H-indazol-3-yl)(methylsulfonyl)amide; Partition coefficient (log P): 5.1; pka: 6.8; Shelf-life: 2 Years; Figure 2) is the active ingredient in Sunlenca [20,21,24].

LEN has three chiral centers and can exist in eight stereoisomeric forms. The (2S,3bS,4aR)-isomer is the main isomeric active ingredient of Sunlenca. Different polymorphs (crystalline, amorphous, solvates, etc.) of LEN are reported. Sunlenca contains crystalline LEN sodium as the active ingredient due to its convenient features, reproducibility, stability (oxidative, hydrolytic, and photolytic), and biopharmaceutical characteristics [20,21].

Sunlenca injection (the sterile and preservative-free clear yellowish-brown solution) contains polyethylene glycol 300 and water for injection as additional non-medicinal ingredients. Sunlenca tablet (beige, capsule-shaped film-coated tablet) also contains many additional non-medicinal ingredients [20,24,25].

### 2.2. Mechanism of Action of LEN

HIV-1 is an enveloped retrovirus. The HIV-1 capsid (a protein shell) contains viral RNA, nucleocapsid, reverse transcriptase, and integrase (Figure 3) [26,27,28]. This HIV-1 capsid contributes to multiple essential processes during different stages of HIV-1 replication, including protecting, transporting, interacting with the host cell, and releasing the viral genome in the host cell (Figure 4) [18,26,29]. A deficiency in the normal function of the HIV-1 capsid impedes different aspects of the HIV-1 life cycle, including the nuclear uptake and integration of viral DNA into the host genome. These features of HIV-1 capsid make it a potential drug target for developing anti-HIV agents [30,31,32].

The protein capsid of HIV-1 comprises repeating subunits called protomers (hexamer). LEN selectively binds at the interface of two hexamer subunits (N74 residue of the N-terminal domain of one hexamer and two residues (N183 and K70) of the C-terminal domain of neighboring hexamer) of HIV-1 capsid [33,34]. This phenomenon causes the development of immature HIV-1 capsid or its inhibition and, ultimately, inhibition of various functions of HIV-1 capsid at different stages of the replication cycle of HIV-1 (Figure 4). Accordingly, LEN interferes at multiple stages of the HIV-1 life cycle, including capsid-mediated nuclear uptake of HIV-1, virus assembly and release, and capsid core formation (Figure 4).

### 2.3. Preclinical Studies

The preclinical studies of LEN are well described in the literature [12,20,25,35,36,37]. The USFDA, Health Canada, and EMA have approved LEN disclosing its complete pharmacological data. Therefore, only a summary of pre-clinical studies is provided in this section. LEN demonstrated an EC50 (half-maximum effective concentration) of 105 pM (MT-4 cells infected with HIV-1), 32 pM (human CD4+ T cells), 20–160 pM (23 clinical isolates of HIV-1), 56 pM (macrophages), and 885 pM (HIV-2 isolates). This data also indicates that LEN is about 8–10 times less active for HIV-2 isolates than HIV-1 [12,35]. The cytotoxicity assay of LEN in human cell lines (MT4, Huh-7, Gal-HepG2, Gal-PC-3, and MRC-5) and primary human cells (hepatocytes, Quiescent PBMCs, Stimulated PBMCs, CD4+ T-lymphocytes, and Monocyte-derived macrophages) revealed LEN’s half-maximal cytotoxic concentration (CC50 in µM) from 24.7 µM to >50 µM with a corresponding selectivity index (CC50/EC50 for HIV-1) of 140,000 to 1,670,000 [35,38,39]. The off-target assay of LEN (10 µM) among 87 different receptors, enzymes, and ion channels did not produce any significant response [20,25,38,39]. These data signified the potency of LEN against HIV-1 and the low potential of LEN to provide off-target effects. The effect of LEN against drug-resistant isolates of HIV-1 was also promising (Table 2) [35].

The pharmacokinetic data of LEN in the animal models (Sprague–Dawley rats, Beagle dogs, and Cynomologous monkeys) is disclosed in the patent/patent application file by Gilead and the documents published by the USFDA, EMA, and Health Canada [20,24,25,36,37]. LEN was tested in dogs (100 mg/kg) for the assessment of its effect on the heart (blood pressure, heart rate, ECG, QT, and QTc) and rats (100 mg/kg and 10 mg/kg) for the assessment of its effect on the central nervous system and respiratory system. These animal-based studies revealed no significant concern for the heart, CNS, and respiratory systems. LEN demonstrated no effects on animal fertility and was non-carcinogenic and non-mutagenic.

### 2.4. Clinical Studies on LEN

We searched the clinical trial database and Gilead’s website using the abovementioned keywords [14,40]. This search provided eight clinical studies on LEN (Table 3).

The study results of three clinical studies (NCT03739866, NCT04143594, and NCT04150068) are available on the clinical trial database [40]. Our PubMed search revealed some clinical studies related to LEN. These studies are summarized below.

The data of the clinical phase 1 study (NCT03739866) is available on the clinical trial website and in the literature [35]. This study demonstrated LEN’s safety, efficacy, and appreciable pharmacokinetic parameters.

The clinical phase 2 data (NCT04143594) of the combination of LEN with other ART has been published [16,41]. A virological suppression of 85–92% was observed after 54 weeks with the combination of LEN with different ARTs (emtricitabine, tenofovir, and bictegravir). The study revealed headache and nausea as the most frequent adverse events with oral treatment, whereas erythema, swelling, and pain were associated with SC administration of LEN.

The clinical phase 3 data of NCT04150068 is recently published [19]. This study was performed on MDR HIV-1 patients. The LEN-treated group displayed a greater reduction of the viral load in >81% of the patient than the placebo group. A commentary on the ongoing clinical phase 3 data of LEN (NCT04925752, Purpose 2 trial) has been publicized [42]. However, this study is silent about the conclusive and full outcomes of the study.

A proof-of-concept clinical trial has also been performed that analyzed the activity of LEN against LEN-associated resistance mutations [43]. This study revealed a beneficial inverse association between the replication capacity of HIV-1 and drug resistance. This study also revealed capsid mutation (Q67H) development with LEN monotherapy in two participants.

### 2.5. Pharmacological Properties of LEN

LEN’s important pharmacological parameters (dosing, pharmacokinetics, adverse effects, warning, toxicity, and drug interactions) are summarized in Table 4.

### 2.6. The Development Cycle of LEN

The important event (patent filing, clinical trials, and drug regulatory affairs) of LEN is depicted in Figure 5.

## 3. Patent Literature

The patent search was conducted on 9 February 2023, using different keywords of LEN (GS-6207, GS-714207, lenacapavir, Sunlenca, and J05-AX31) on different patent databases, including USPTO, Espacenet, and Patent scope [45,46,47,48]. When the USFDA approves a drug, the innovator also provides a list of relevant patents related to the approved drug products to the USFDA. The USFDA lists these patents in the Orange Book Database (OBD) [49,50]. The OBD was also searched to get information about the relevant patents related to Sunlenca. The patents/patent applications that specifically mentioned LEN-based inventions in their claim directly or indirectly were segregated. The redundant/duplicate patents/patent applications were removed, and the remaining patent documents are summarized in Table 5.

## 4. Discussion

HIV/AIDS-related morbidity and mortality have decreased over the last two decades. However, the MDR HIV-1 infection remains an unmet medical need [64]. Most anti-HIV regimens require daily dosing and cause patients non-compliance and non-adherence to the dosing schedule [44,60,65]. Many anti-HIV drugs have similar chemical structures (NRTI and NNRTI). This structural similarity is a common cause of cross-resistance among these drugs. HIV-1 develops broad multi-class drug resistance due to mutations in HIV-1, non-adherence to the prescribed treatment, and cross-resistance of anti-HIV drugs [36]. This phenomenon is seen in heavily treatment-experienced HIV-1 infected patients and limits the effectiveness of ART [26]. Accordingly, these patients have limited treatment options and are at a higher risk of morbidity and mortality [3]. Long-acting anti-HIV drugs can improve the quality of life of this class of HIV-1 patients and can combat the MDR HIV-1 infection [44]. Accordingly, the scientific fraternity is trying to develop long-acting anti-HIV agents for this class of patients.

LEN is a long-acting and patient compliant (oral/subcutaneous administration and less frequent dosing schedule) first-in-class HIV-1 capsid inhibitor approved by the USFDA, Health Canada, and EMA for the treatment of MDR HIV-1 infection (Table 1) [11,12,13]. LEN demonstrated a low level of drug-related mutations, is active against MDR HIV-1 infection and does not demonstrate cross-resistance to ARTs due to its novel mechanism of action (Figure 4) [3,66,67]. LEN is also an excellent drug for patients having difficult or limited access to healthcare facilities [68,69]. The long-acting and patient-compliant LEN can address the unmet need of MDR HIV-1, high-risk population of HIV-1, and HIV-1 infected patients with a history of ineffective diverse treatment, drug resistance, and viremia level [13].

HIV-1 variants (Q67H, N74D, and Q67H/N74D) with mutations in the binding sites of LEN have been identified [20,70]. One report established that certain mutations do not affect LEN’s in vitro efficacy [71]. Another report states a decrease in the anti-HIV activity of LEN against some variants (M66I, K70H, and Q67H + K70R). The exact effects of these mutilations on the efficacy of LEN are yet to be explored and need further investigations.

HIV-1 is the causative agent for AIDS, which makes them highly susceptible to fatal and opportunistic infections and other diseases, including opportunistic infections such as tuberculosis (TB), hypertension, hepatitis, lipidemia, diabetes, asthma, and mental health issues [72,73,74,75]. The accompanied diseases make HIV treatment complex and warrant drug interaction studies (drug–drug, drug-food, and drug-disease interaction) [69]. Some LEN-based drug interaction studies have been performed (Table 3). However, there remain a lot of new drug interaction studies for LEN (Figure 6). This is one of the interesting areas for scientists to work upon. It is also imperative to note that if LEN therapy is discontinued, it may remain in the patient’s systemic circulation for long periods. Therefore, drug interaction must be handled before starting the new anti-HIV treatment [20,24,25].

Monotherapy with an anti-HIV agent provides temporary effects, and soon HIV develops resistance to monotherapy. Accordingly, combination therapy for HIV-1 treatment is recommended [13,60,65,69,76]. Cabenuva (rilpivirine + raltegravir) is the first long-acting FDA-approved therapy for HIV-1 infections [44]. One study has demonstrated the additive/synergistic effects of combining LEN with rilpivirine, cabotegravir, and islatravir [44]. The scientists are also trying to develop a suitable long-acting partner for LEN [13]. Some traditional medicines, vitamins, and immunity boosters help treat HIV-1 infection [77,78]. Combining LEN with these supplements may also provide a patient-compliant LEN-based therapy.

The patent literature of LEN reveals its inventions related to compound per se, manufacturing process, crystalline/amorphous polymorphs, method of treating HIV infection, and drug combinations with some drugs/vaccines (islatravir, bictegravir, tenofovir, abacavir, protease inhibitor, NNRTI, NRTI, integrase inhibitor, and pharmacokinetic enhancers) (Table 4). However, most of these patents/applications are silent about the experimental evidence of the claimed inventions. This knowledge gap creates an opportunity to explore this area of HIV-1 treatment. There is also scope for developing more inventions related to new LEN-based regimens with improved treatment outcomes, drug combinations with specific drugs (ART, anti-TB agent, and pharmacokinetic modulators), indications for other viral diseases, dosage forms, analog with anti-HIV activity, co-crystals and salts with improved pharmaceutical properties, economical/simple manufacturing process, and particle size. LEN is a newly approved anti-HIV agent, and LEN-based therapy requires additional monitoring. This step will quickly recognize new safety data. All this additional research may provide more long-acting, potent, non-toxic, stable, and patient-compliant LEN-based treatment with favorable pharmacokinetic parameters for MDR HIV-1 infections and associated opportunistic infections such as TB.

## 5. Conclusions

LEN inhibits the HIV-1 replication cycle at different stages due to its unique HIV-1 capsid inhibitory mechanism, making it active against MDR HIV-1 infection. Introducing long-acting and patient-compliant LEN therapy with no known cross-resistance and drug resistance brings new hope for MDR HIV-1 infected patients. The patented long-acting LEN-based combination therapies with islatravir, bictegravir, and tenofovir are in the clinical trial. The authors foresee the possibility of developing many LEN combinations with existing ARTs for HIV-1 infection treatment. Opportunistic infections such as TB accompany AIDS. Therefore, the drug–drug interaction-based combination therapy of LEN and anti-TB drugs also need investigations. The marketing approval of LEN is a milestone for HIV-1-infected patients, and it would be interesting to see the development of several advantageous LEN-based therapies for MDR HIV-1 infection.

## Figures and Tables

**Figure 1 medicina-59-01041-f001:**
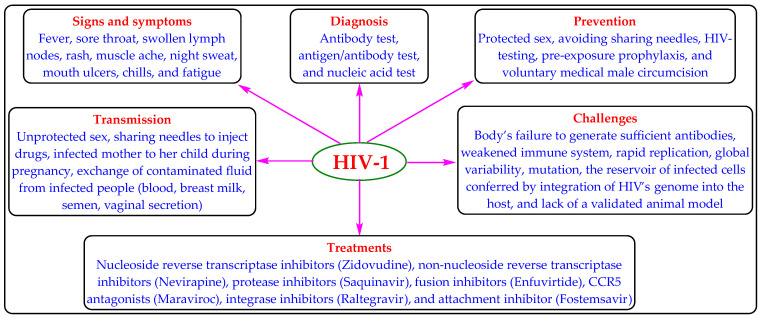
The mode of transmission, symptoms, diagnosis, treatment, prevention, and challenges of HIV-1 infection.

**Figure 2 medicina-59-01041-f002:**
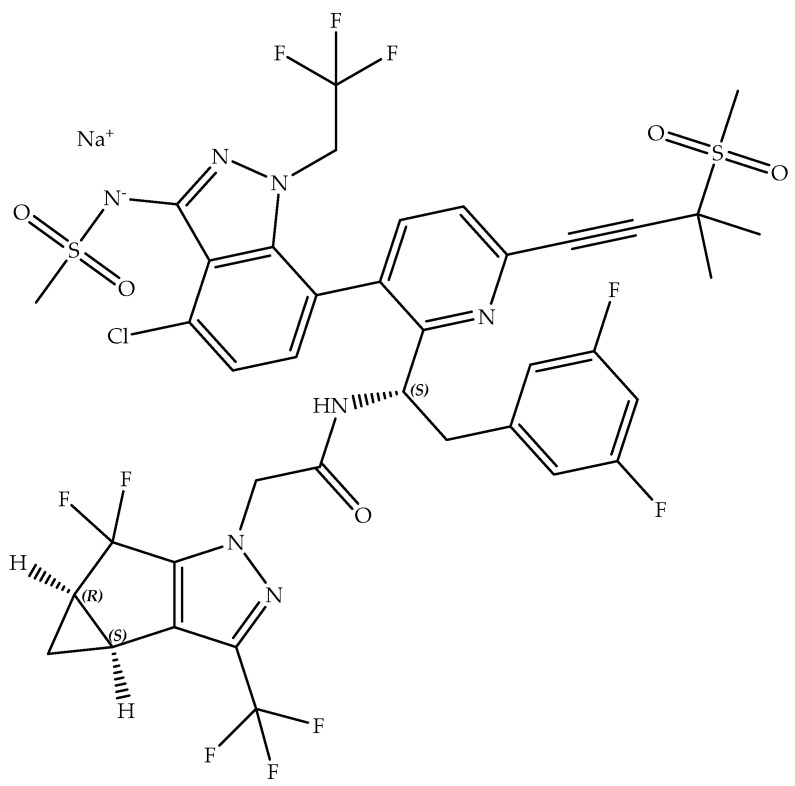
Chemical structure of Lenacapavir sodium.

**Figure 3 medicina-59-01041-f003:**
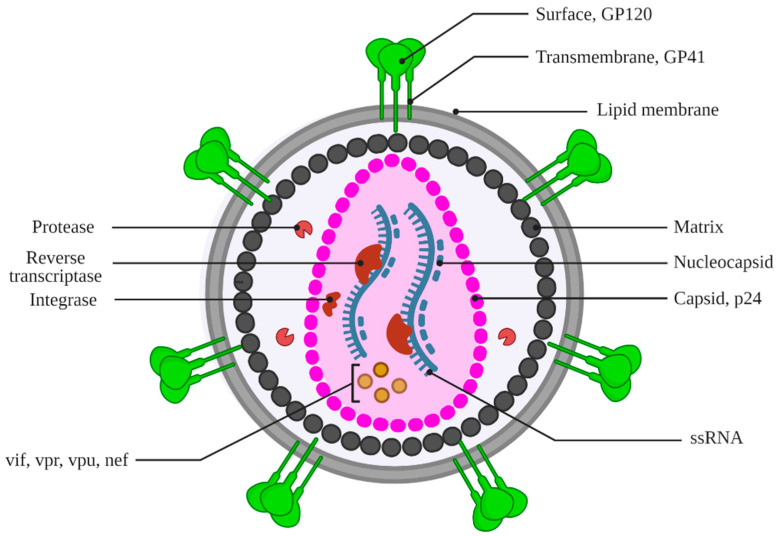
Structure of HIV-1 virion. Image created with Biorender.com.

**Figure 4 medicina-59-01041-f004:**
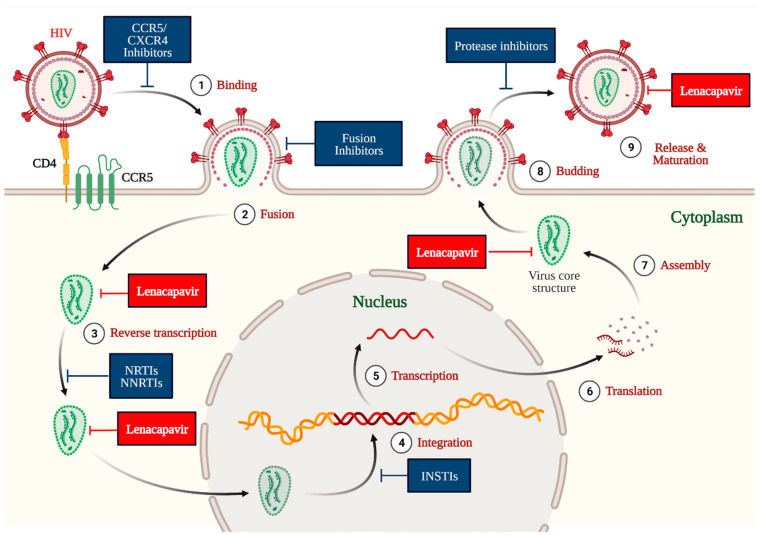
The mechanism of action of LEN at the multiple stages of the HIV-1 life cycle. Image created with Biorender.com.

**Figure 5 medicina-59-01041-f005:**
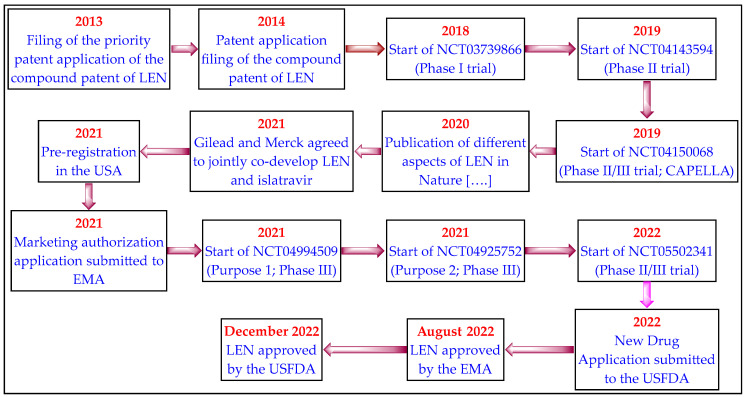
The development cycle of LEN.

**Figure 6 medicina-59-01041-f006:**
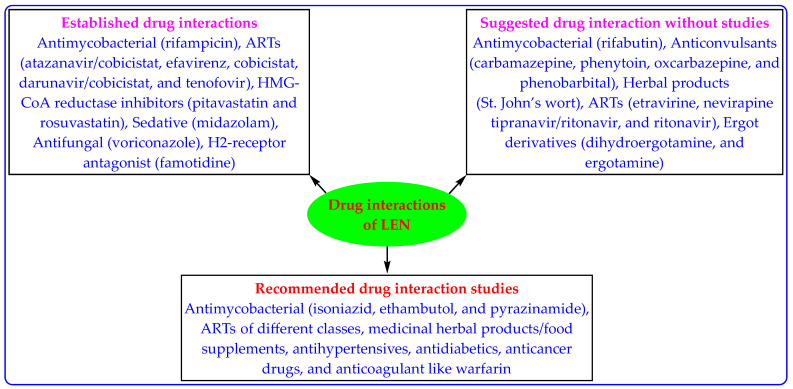
Reported and recommended LEN-based drug interaction studies.

**Table 1 medicina-59-01041-t001:** Product details of Sunlenca.

Active Pharmaceutical Ingredient (Proprietary Name; Applicant)	Dosage Form (Route; Strength)	Approval Date (Marketing Status)	Indication
Lenacapavir Sodium (Sunlenca; Gilead Sciences)	Solution for injection (Subcutaneous; EQ 463.5 mg base/1.5 mL and EQ 309 mg base/mL)	22 December 2022, by the USFDA; 1 November 2022, by Health Canada; 17 August 2022, by the EMA (Prescription)	In combination with other ARTs, for the treatment of heavily treatment-experienced patients with previous failed ART due to MDR HIV-1 infection, intolerance, or safety issues
	Tablet (Oral; EQ 300MG BASE)		

**Table 2 medicina-59-01041-t002:** The activity of LEN against drug-resistant isolates of HIV-1 [35].

ART Class	HIV-1 Mutant	Fold Resistance (Mean ± Standard Deviation)		
		LEN	Control	ART Agent
**Nucleoside reverse transcriptase inhibitor**	K65R	0.6 ± 0.2	Emtricitabine	14.1 ± 2.6
	M184V	0.3 ± 0.3	Emtricitabine	>23.0
	6TAMs	0.2 ± 0.1	Emtricitabine	4.0 ± 2.8
**Non-nucleoside reverse transcriptase inhibitor**	Y188L	0.5 ± 0.1	Efavirenz	>22.5
	L100I + K103N	0.5 ± 0.2	Efavirenz	>22.5
	K103N + Y181C	0.6 ± 0.2	Efavirenz	>22.5
**Integrase inhibitor**	E138K + Q148K	0.6 ± 0.3	Elvitegravir	>53.8
	G140S + Q148R	0.8 ± 0.3	Elvitegravir	>53.8
	E92Q + N155H	0.8 ± 0.4	Elvitegravir	>53.8
	N155H + Q148R	1.2 ± 0.7	Elvitegravir	>52.9
**Maturation inhibitor**	V230I in Capsid	0.7 ± 0.2	Bevirimat	>67.5
	V7A in 14-amino acid spacer peptide 1 (SP1)	0.8 ± 0.4	Bevirimat	>67.5
**Protease inhibitor**	M46I + I50V	0.7 ± 0.2	Darunavir	27.1 ± 23.1
	I84V + L90M	0.3 ± 0.1	Atazanavir	32.7 ± 7.8
	G48V + V82A + L90M	0.5 ± 0.2	Atazanavir	31.0 ± 11.9
	G48V + V82S	0.4 ± 0.2	Atazanavir	15.2 ± 3.2

**Table 3 medicina-59-01041-t003:** A summary of the Gilead-sponsored interventional clinical studies on LEN for the treatment of HIV infection.

NCT Number (Other IDs; Start Date; Completion Date; Last Update)	Objective/Interventions (Number Enrolled; Allocation; Primary Purpose)	Phase (Status; Location; Results)
**NCT05502341** (GS-US-621-6289 and 2022-500929-33-00; 16 August 2022; January 2026; 18 January 2023)	Safety and efficacy of Bictegravir/LEN versus ART (671; Randomized; Treatment)	2/3 (Recruiting; United States; Not available)
**NCT05052996** (GS-US-563-6041; 5 October 2021; August 2023; 21 December 2022)	Safety and efficacy of the combination of LEN and islatravir (136; Randomized; Treatment)	2 (Active; United States; Not available)
**NCT04994509** (GS-US-412-5624 and DOH-27-072021-6125; 30 August 2021; March 2024; 18 January 2023)	Safety and efficacy of LEN and emtricitabine/tenofovir alafenamide (F/TAF) for PrEP—PURPOSE 1 (5010; Randomized; Prevention)	3 (Recruiting; South Africa; Not available)
**NCT04925752** (GS-US-528-9023 and DOH-27-102021-6681; 28 June 2021; January 2024; 31 January 2023)	Safety and efficacy of LEN in preventing HIV-1 infection—PURPOSE 2 (3000; Randomized; Treatment)	3 (Recruiting; United States; Not available)
**NCT04811040** (GS-US-536-5816; 8 April 2021; 9 June 2022; 8 November 2022)	Safety and tolerability of the combination of LEN with teropavimab and zinlirvimab (32; Randomized; Treatment)	1 (Active; United States; Not available)
**NCT04150068** (GS-US-200-4625 and 2019-003814-16; 21 November 2019; 5 October 2020; 19 October 2022)	Safety and efficacy of LEN as an add-on to a failing HIV-1 therapy due to drug resistance—CAPELLA (72; Randomized; Treatment)	2/3 (Active; United States; Results posted on 20 October 2021)
**NCT04143594** (GS-US-200-4334; 22 November 2019; 30 September 2021; 19 December 2022)	Safety and efficacy of regimens (LEN + ARTs) against HIV-1—CALIBRATE (183; Randomized; Treatment)	2 (Active; United States; Results posted on 19 December 2022)
**NCT03739866** (GS-US-200-4072; 26 November 2018; 15 June 2020; 9 April 2021)	Safety, antiviral activity and pharmacokinetic study of LEN in HIV-1 infected patients (53; Randomized; Treatment)	1 (Completed; United States; Results posted on 9 December 2020)

**Table 4 medicina-59-01041-t004:** Important pharmacological parameters of LEN.

Parameter	Summary
**Dose/Regimen**	**Option 1** Day 1: Abdominal SC injection (2 injections of 463.5 mg/1.5 mL) and Tablet (2 × 300 mg); Day 2: Tablet (2 × 300 mg); Maintenance dose every six months: Abdominal SC injection (2 injections of 463.5 mg/1.5 mL) [25]
	**Option 2** Day 1: Tablet (2 × 300 mg); Day 2: Tablet (2 × 300 mg); Day 8: Tablet (2 × 300 mg); Day 15: Abdominal SC injection (2 injections of 463.5 mg/1.5 mL); Maintenance dose every six months: Abdominal SC injection (2 injections of 463.5 mg/1.5 mL) [25]
**Absorption**	Absolute bioavailability: 6–10% (oral) and 100% (SC); T_max_: 4 h (oral) and 77 to 84 days (SC) [3,12,20,24,25]
**Volume of distribution**	19240 (oral); 9500-11700 (SC); 976 Litres
**Protein binding**	>98.5% [20,24,25]
**Metabolism**	CYP3A and UGT1A1 metabolize LEN. LEN does not induce CYP3A4. LEN is neither a substrate nor induces/inhibits CYP2D6, CYP2C19, CYP2C9, CYP2C8, CYP2B6, and CYP1A2. LEN does not inhibit UGT1A1 and anion transporters [12,20,24,25].
**Route of elimination**	Excretion of the unchanged drug into feces; <1% is excreted in urine [20,24,25].
**Half-life**	Oral: 10–12 days; Subcutaneous: 8–12 weeks [20,24,25].
**Clearance**	Fifty-five days (oral) and 4.2 weeks (SC) [20,24,25].
**Adverse Effects**	Nausea and reactions at the injection site (swelling, pain, erythema, nodule, induration, pruritus, extravasation, or mass) are common. The uncommon but possible adverse reactions include immune reconstitution syndrome, proteinuria, hyperglycemia, glycosuria, and increased creatinine and liver enzymes [12,20,24,25].
**Drug interactions**	Coadministration of some drugs (carbamazepine, oxcarbazepine phenobarbital, phenytoin, efavirenz, nevirapine, rifabutin, rifampin, and rifapentine) and strong CYP3A inducers may cause sub-therapeutic effects of LEN may cause the development of drug resistance. No significant interaction reported with some drugs (darunavir + cobicistat, cobicistat, famotidine, pitavastatin, rosuvastatin, tenofovir, alafenamide, and voriconazole) [20,24,25,44].
**Food Interactions**	The tablet may be taken with or without food [20,24,25].
**Contraindications**	Concomitant administration of Sunlenca with strong CYP3A inducers/inhibitors, atazanavir/cobicistat, and some herbal products (St. John’s wort) is not recommended [12,20,24,25].
**Warning/Precautions**	Immune reconstitution syndrome and autoimmune disorders such as Grave’s disease, autoimmune hepatitis, and polymyositis. Non-adherence to the recommended dose can cause drug resistance [20,24,25].
**Toxicity/Overdose**	Limited data is available on the toxicity of LEN. Toxicity treatment is based on signs and symptoms, along with supportive care. Dialysis may not be effective because LEN is a highly protein-bound drug. No cardiotoxicity is reported using LEN [20,24,25].
**Special population**	No study has been carried out on the effect of LEN among the elderly, pediatric, and pregnant/breastfeeding women population [20,24,25].

**Table 5 medicina-59-01041-t005:** Summary of the LEN-based patents/patent applications.

Patent/Patent Application (Applicant; Filing Date)	Summary
**OBD-Listed Patents and Their Family Members**	
**US9951043B2** (Gilead Sciences; 28 February 2014)	This patent (priority application date of 1 March 2013) has an estimated expiry date of February 28, 2034, which may further be extended based on USFDA laws. This patent relates to therapeutic compounds for treating HIV infection and generically claims LEN and its pharmaceutically acceptable salts [51].
**US10071985B2** (Gilead Sciences; 17 August 2017)	This patent (priority application date of 19 August 2016) has an estimated expiry date of 17 August 2037, which may further be extended based on USFDA laws. This patent specifically claims LEN, its salts, pharmaceutical compositions, and amorphous solid form. This patent also exemplifies and provides the data of the antiviral assay in MT4 cells, cytotoxicity (EC50 and CC50), the pharmacokinetic study by IV route (Cmax, AUC, half-life, distribution, and clearance) in Sprague–Dawley rats, Beagle dogs, and Cynomologous monkey and the metabolic stability (cultured human liver hepatocytes) of LEN. The patent specifications state LEN is a potent antiviral compound with improved pharmacokinetic parameters [37].
**US11267799B2** (Gilead Sciences; 16 August 2018)	This patent (priority application date of 17 August 2017) has an estimated expiry date of 16 August 2038, which may further be extended based on USFDA laws. This patent claims crystalline Form I (the active ingredient of Sunlenca), Form II, and Form III of LEN sodium salt. It also claims the pharmaceutical composition of the claimed crystalline forms to treat/prevent HIV infection. Its specification also mentions amorphous and crystalline Form IV of LEN [52].
**US10654827B2** (Gilead Sciences; 25 June 2018)	This patent (priority application date of 19 August 2016; a family member of US10071985B2) has an estimated expiry date of 17 August 2037. This patent claims a method of treating HIV infection with LEN or its salts. It also claims treatment of HIV infection with a combination of LEN and existing HIV treatments (protease inhibitors, NRTIs, NNRTIs, integrase inhibitors, entry inhibitors, maturation inhibitors, and capsid inhibitors) and other therapies (immune-based therapies, PI3K inhibitors, HIV antibodies, HIV p17 matrix protein inhibitors, HIV vif gene modulators, HIV-1 viral infectivity factor inhibitors, HIV-1 Nef modulators, HIV-1 splicing inhibitors, HIV ribonuclease inhibitors, HIV GAG protein inhibitors, HIV POL protein inhibitors, reverse transcriptase priming complex inhibitors, HIV gene therapy, pharmacokinetic enhancers, and HIV vaccines). This patent specifically claims to treat HIV-1 infection with LEN in a combination of 4′-ethynyl-2-fluoro-2′-deoxyadenosine, bictegravir, tenofovir, or abacavir [53].
**US2021009555A1** (Gilead Sciences; 26 May 2020)	This patent application (Status: Notice of Allowance mailed) is a family member of US9951043B2 and claims a process for preparing LEN [54].
**US2018273508A1** (Gilead Sciences; 2 March 2018)	This patent application (Status: Abandoned) is a family member of US9951043B2 and claims a process for preparing LEN [55].
**US2014303164A1** (Gilead Sciences; 28 February 2014)	This patent application (Status: Abandoned) is a family member of US9951043B2 and claims LEN-related compounds [56].
**US2020262815A1** (Gilead Sciences; 25 November 2019)	This patent application (Status: Notice of Allowance mailed) is a family member of US10071985B2 and claims compounds that can be used to prepare LEN.
**US2022251069A1** (Gilead Sciences; 26 January 2022)	This patent application (Status: Under examination) is a family member of US11267799B2 and claims the use of polymorphs of LEN in combination with other antivirals, including 4′-ethynyl-2-fluoro-2′-deoxyadenosine, bictegravir, tenofovir or abacavir to treat HIV infection [57].
**US2020038389A1** (Gilead Sciences; 15 July 2019)	A method of treating HIV-1 infection in a heavily treatment-experienced patient (e.g., patients with MDR HIV infection) with a therapeutically amount of LEN or its pharmaceutically acceptable salt alone or in combination with a protease inhibitor, NNRTI, NRTI, integrase inhibitor, and pharmacokinetic enhancers [36].
**Other important patents/applications of LEN**	
**WO2022159877A1** (Brii Biosciences; 25 January 2022)	A combination of capsid inhibitors (LEN) and an adenosine derivative (islatravir. an NRTI) for the treatment/prevention of HIV infection. This patent application is silent about the experimental details of the combination of LEN and islatravir [58].
**WO2022155258A1** (Gilead Sciences; 12 January 2022)	A vaccine for HIV-1 infection comprising polypeptide segments of one or more HIV-1 proteins that can be co-administered with anti-HIV agents, including capsid inhibitors (LEN, GS-CA1, AVI-621, AVI-101, AVI-201, and AVI-301). This patent application is silent about the experimental details of the combination of LEN and the claimed vaccine [59].
**WO2021236944A1** (Gilead Sciences; 20 May 2021)	A method of treating/preventing HIV infection utilizing bictegravir (injection) alone or in combination with a capsid inhibitor (LEN. GS-CA1, AVI-621, AVI-101, AVI-201, and AVI-301) or other anti-HIV drugs. This patent application is silent about the experimental details of the combination of LEN and bictegravir [60].
**KR20220112771A** (Viiv Healthcare; 7 December 2020)	A method of treating HIV infection with a combination of cabotegravir and a capsid inhibitor (LEN) or other anti-HIV agents. This patent application is silent about the experimental details of the combination of LEN and cabotegravir [61].
**WO2021011891A1** (Gilead Sciences; 17 July 2020)	A method of treating HIV infection with a long-acting pharmaceutical formulation of tenofovir alafenamide containing sucrose acetate isobutyrate, which may optionally contain a capsid inhibitor (LEN). This patent application is silent about the experimental details of the combination of LEN and tenofovir [62].
**WO2020247933A2** (Univ Missouri; 8 June 2020)	This patent application claims cyclic or linear peptides as HIV-1 capsid inhibitors [63].

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
