# Peer review of "Highlights on the Development, Related Patents, and Prospects of Lenacapavir: The First-in-Class HIV-1 Capsid Inhibitor for the Treatment of Multi-Drug-Resistant HIV-1 Infection"

_medicina, 2023, doi:10.3390/medicina59061041_

Round 1

Reviewer 1 Report

Dear Authors,

You present here a review concerning the development, the related patents and the prospects for Lenacapavir, a new capsid inhibitor, to be used in the treatment of MDR HIV-1 infection.

The subject of anti-HIV drug development is important and your manuscript might be interesting for the readers if it would be more organised. For that, I have some suggestions:

1. change the way of data presenting in the tables, especially Tables 4 and 5. The information is crowded and difficult to follow.

2. in the molecular formula, the numbers should be written subscript.

3. in figure 6, you wrote "efavirenzb" and "cobicistatb". What does the "b" stand for?

4. considering that your work is a review, I suggest you have a more detailed "Conclusions" part.

The references used are well chosen and presented.

Dear Authors,

The quality of English used in the writing of your review is fine.

Author Response

Please see our attached responses.

Reviewer 2 Report

The authors have presented a comprehensive review of the newest HIV-1 capsid inhibitor Lenacapavir. The paper gives an insight into the mechanism of action of Lenacapavir, as well as the results of preclinical and clinical trials. Data on the pharmacokinetics and pharmaceutical properties of the new drug are also presented. An overview of the recent patent literature is provided, including data from 2023. Overall, the review is quite good, written in good language, easy to read and well illustrated and the topics are exceptionally relevant.

Minor comments.

1) Line 99 "(2S,3bS,4aR)-isomer" -

a search of the name Lenacapavir gives the following variants of the IUPAC name - N-[(1R)-1-[3-[4-Chloro-3-(cyclopropylsulfonylamino)-1-(2, 2-difluoroethyl)indazol-7-yl]-6-(3-methyl-3-methylsulfonylbut-1-ynyl)pyridin-2-yl]-2-(3,5-difluorophenyl)ethyl]-2-[(2R,4S)-9-(difluoromethyl)-5,5-difluoro-7,8-diazatricyclo[4.3.0.02,4]nona-1(6),8-dien-7-yl]acetamide, or

N-((S)-1-(3-(4-chloro-3-(methylsulfonamido)-1-(2,2,2-trifluoroethyl)-1H-indazol-7-yl)-6-(3-methyl-3-(methylsulfonyl)but-1-yn-1-yl)pyridin-2-yl)-2-(3, 5-difluorophenyl)ethyl)-2-((3bS,4aR)-5,5-difluoro-3-(trifluoromethyl)-3b,4,4a,5-tetrahydro-1H-cyclopropa[3,4]cyclopenta[1,2-c]pyrazol-1-yl)acetamide. 

I suggest the authors to give the numbering of atoms of the target molecule in Figure 2 to avoid confusion.

2). At the time the authors submitted their paper to Medicina, another review on Lenacapavir had appeared - Tailor, M. W., Chahine, E. B., Koren, D., & Sherman, E. M. (2023). Lenacapavir: A Novel Long-Acting Capsid Inhibitor for HIV. The Annals of pharmacotherapy, 10600280231171375. Advance online publication. https://doi.org/10.1177/10600280231171375

Although the review by Tafadzwa Dzinamarira and colleagues seems to be more thorough, I think it is important to note the fact that there are parallel published review papers in this area.

Author Response

Please see our attached responses. 
